# Effect of the Administration of *Cordyceps militaris* Mycelium Extract on Blood Markers for Anemia in Long-Distance Runners

**DOI:** 10.3390/nu16121835

**Published:** 2024-06-11

**Authors:** Akira Nakamura, Eri Shinozaki, Yoshio Suzuki, Kazuki Santa, Yoshio Kumazawa, Fumio Kobayashi, Isao Nagaoka, Natsue Koikawa

**Affiliations:** 1Faculty of Health and Sports Science, Juntendo University, Chiba 270-1695, Japan; aknaka@juntendo.ac.jp; 2Koyama Memorial Hospital, Ibaraki 314-0030, Japan; 3Graduate School of Health and Sports Science, Juntendo University, Chiba 270-1695, Japan; yssuzuki@juntendo.ac.jp (Y.S.); nkoikawa@juntendo.ac.jp (N.K.); 4Department of Biotechnology, Tokyo College of Biotechnology, Tokyo 114-0032, Japan; k-santa@juntendo.ac.jp; 5Faculty of Medical Science, Juntendo University, Chiba 279-0013, Japan; 6Department of Biochemistry, School of Medicine, Juntendo University, Tokyo 113-8421, Japan; kumazawa@vsj.co.jp; 7Vino Science Japan Inc., Kanagawa 210-0855, Japan; 8La Vie En Sante Ltd., Ibaraki 315-0014, Japan; f_kobayashi@nifty.com; 9Department of Biochemistry and Systems Biomedicine, Graduate School of Medicine, Juntendo University, Tokyo 113-8421, Japan

**Keywords:** Dongchong-hongxiacao, long-distance runner, anemic marker, ferritin, hemoglobin, hematocrit, creatine kinase

## Abstract

In the present study, we conducted a placebo-controlled, double-blind, parallel-group comparison trial in which an extract of *Cordyceps militaris* (CM) mycelium was administered to long-distance runners for 16 weeks during the pre-season training period and blood test markers for anemia were investigated. The results indicated that the change rates of serum ferritin levels were moderately increased in the CM group (*n* = 11) but decreased in the placebo group (*n* = 11) during the study period, and the levels were significantly increased in the CM group compared with those in the placebo group at 4 weeks and 8 weeks after the test food intake (*p* < 0.05). Moreover, the change rates of hemoglobin and hematocrit were significantly increased in the CM group compared with those in the placebo group at 8 weeks after the test food intake (*p* < 0.05). These observations suggest that the intake of test food containing *Cordyceps militaris* mycelium extract is expected to effectively maintain the hemoglobin and hematocrit levels in long-distance runners, possibly via the suppression of the decrease in iron storage, which is reflected by serum ferritin, during pre-season training. Furthermore, the levels of creatine kinase were increased above the normal range in both the placebo and CM groups at registration. Interestingly, the creatine kinase levels were significantly decreased in the CM group compared with those in the placebo group at 16 weeks after the test food intake (*p* < 0.05). These results suggest that *Cordyceps militaris* mycelium extract exhibits a protective action on the muscle damage observed in long-distance runners and may suppress muscle injury. Together, these observations suggest that *Cordyceps militaris* mycelium extract exhibits an improving effect on the markers for not only anemia, but also muscle injury in long-distance runners during pre-season training.

## 1. Introduction

*Ophiocordyceps sinensis*, also known as *Cordyceps sinensis* or Dong Chong Xia Cao (winter worm summer grass), is an entomopathogenic fungus with a club-shaped fruiting body originally found in the alpine region of China at an altitude of 3000–5000 m [1,2,3,4]. The host of the fungus is the larvae of the Chinese moth (*Hepialus armoricanns* Oben). *Ophiocordyceps sinensis* infects the larva in winter, slowly grows inside the larva, utilizes the nutrients of the larva and emerges from the ground in summer with the distinctive shape of “Dongchong-hongxiacao”, retaining the larva in the ground.

Since the Chinese moth (Oben) that is susceptible to *Ophiocordyceps sinensis* does not naturally live in Japan, a variety of fungi including *Ophiocordyceps sobolifera*, which grows in the larva of *Platypleura kaempferi*, and *Cordyceps militaris*, which grows in the larvae of the moths, have been used for Dongchong-hongxiacao [4,5].

Some research has already been conducted on the effect of a supplement containing *Cordyceps sinensis* on exercise performance. In one study, 10 healthy untrained males were administered with an extract of *Cordyceps sinensis* mycelium for one week and subjected to a treadmill exercise load test. The results indicated that, with the administration of the *Cordyceps sinensis* mycelium extract, the blood glucose level was maintained during exercise, whereas the blood lactic acid level was decreased, and the increase in blood sugar was suppressed after the exercise [6]. Furthermore, 18 male long-distance runners were administered with *Cordyceps sinensis*–containing supplement for two weeks during high-altitude training. Interestingly, with the administration, the treadmill exercise times were significantly prolonged and aerobic capacity was increased in the runners [7]. Moreover, 14 male middle- and long-distance runners were administered with *Cordyceps sinensis* extract for two months, and the results indicated that reticulocytes were increased after the administration [8]. These observations suggest that the administration of *Cordyceps sinensis* extract may enhance aerobic capacity in athletes, possibly by increasing reticulocytes and improving oxygen transport. However, the values of red blood cell count, hematocrit and hemoglobin were not significantly changed with the administration of *Cordyceps sinensis* extract in the study by Chen et al. [7].

During the pre-season training that is performed before the official season, endurance runners are likely to develop sports anemia because of iron loss, which is associated with sweating and foot-strike hemolysis [9,10]. Based on the above findings, we hypothesized that the administration of Dongchong-hongxiacao may ameliorate anemia observed in endurance runners during pre-season training. Thus, in the present study, we conducted a placebo-controlled, double-blind, parallel-group comparison trial in which the extract of *Cordyceps militaris* mycelium was administered to long-distance runners for 16 weeks during the pre-season training period, and the blood test markers for anemia were investigated. We utilized extracts of *Cordyceps militaris* mycelium instead of *Cordyceps sinensis* extracts, although the effects of *Dongchong-hongxiacao* on exercise performance have been previously evaluated using supplements containing *Cordyceps sinensis* [6,7,8]. This is because the Chinese moth (Oben) that is susceptible to *Cordyceps sinensis* does not naturally live in Japan, so extracts of other fungi that infect and grow in the larvae of insects, including *Cordyceps militaris*, are used for Dongchong-hongxiacao in Japan [4,5].

## 2. Materials and Methods

### 2.1. Subjects

A total of 22 healthy male long-distance runners participated in this study. The inclusion criteria included male long-distance runners aged from 18 to 24 years old belonging to the Faculty of Health and Sports Science, Juntendo University (Chiba, Japan). The exclusion criteria included individuals with significant anxiety about the intake of test foods, which could disturb their normal daily life; individuals suffering from diseases to be treated, including anemia (judged by a preliminary blood test); individuals taking medicines (including iron supplements) prescribed by a physician during the test period; individuals who donated a blood component or 200 mL of whole blood within one month prior to the start of this study; individuals who donated 400 mL of blood in the four months prior to the start of this study; individuals whose blood collection volume exceeded 800 mL in the previous 12 months, including the blood collection volume in this study; individuals who were participating in other trials or had completed a similar trial in the previous four weeks; individuals taking medication or supplements aimed at improving anemia; individuals with anemia-related diseases or conditions; and individuals who were judged to be inappropriate for participating in this study by the principal investigator.

Participants who responded to the recruitment for the study were provided with a research consent document and a thorough explanation of the research content. They were required to submit written voluntary consent to participate in the study. In addition, if the participants were minors, written consents were obtained from their guardians.

All 22 subjects completed the test food intake and measurements; however, two subjects (one each from the placebo group and the *Cordyceps militaris* (CM) group) did not participate in the 5000 m run conducted after the study period, so they were excluded from the 5000 m record. Since there were no defects in other measured values, all 22 subjects were included in the analysis (Figure 1). The age of the subjects at the time of registration was 19.7 ± 0.9 years in the placebo group and 20.2 ± 1.1 years in the CM group, and the height of subjects was 170.6 ± 6.6 cm in the placebo group and 172.0 ± 5.4 cm in the CM group. Table 1 shows the weight, body mass index (BMI), body fat percentage, skeletal muscle percentage and basal metabolic rate of the subjects during the study period (0 week at registration, 4 weeks, 8 weeks and 16 weeks after the intake of test foods).

This study was reviewed and approved by the Research Ethics Committee of the Faculty of Sports and Health Sciences, Juntendo University (Juntendo University Ethics Committee No. 31-5; Date of approval, 18 July 2019), and was conducted in accordance with the Declaration of Helsinki.

### 2.2. Research Protocol

This study was a placebo-controlled, double-blind, parallel-group comparison trial. Based on the results of the measurements performed at registration (18 July 2019), the subjects were assigned to the placebo group or the CM group to ensure that there were no significant differences in ages, levels of hemoglobin and erythropoietin, or seasonal best scores in 5000 m run. Thereafter, they started ingesting the placebo or the CM food for 16 weeks from 2 August 2019 to 22 November 2019. All subjects were instructed to take the placebo or the CM food (6 capsules per day; 3 capsules in the morning and 3 capsules in the evening) with water. In addition, if the subjects forgot to take the placebo or the CM food on one day, they were instructed not to take 2 days of doses on the following day. During the study period, they were allowed to eat and exercise as usual, and were prohibited from taking any medications or supplements intended to improve anemia. The placebo and the CM foods were distributed every month, and at the same time, the empty food bags were collected to check the intake status.

During the study period, the subjects recorded their intake of the placebo or the CM food in a diary. The diaries were distributed every month, and the diaries for the previous month were collected at the same time. Adherence to the intervention was evaluated on the basis of the consumption record in the study diary, and <80% adherence was considered a protocol violation. In this study, the adherence was >99% in both the placebo and CM groups.

In addition to the measurement performed at registration (0 week), blood tests and physical measurements were taken at 4 weeks, 8 weeks and 16 weeks after the test food intake (2 August 2019). However, height was measured only at registration. Body weight, body mass index (BMI), body fat percentage, skeletal muscle percentage and basal metabolic rate were measured during the study period using the Omron HBF-208IT body weight and body composition meter (Omron Healthcare Co., Ltd., Kyoto, Japan) based on the impedance method.

A dietary survey using a brief-type self-administered diet history questionnaire (BDHQ) was conducted at registration, and at 4 weeks, 8 weeks, 12 weeks and 16 weeks after the test food intake. BDHQ data was analyzed by the DHQ Support Center (Gender Medical Research Co., Ltd., Tokyo, Japan). In addition, official records for the 5000 m run were measured at registration (July 2019) and after finishing the test food intake (November 2019).

### 2.3. Test Foods

The test food containing *Cordyceps militaris* was produced by La Vie En Sante Ltd. (Ibaraki, Japan) as a *Cordyceps militaris* mycelium extraction powder (patent registered strain of KT165514). This extraction powder is a food product that has passed the anti-doping certification program (INFORMED-CHOICE) administered by LGC Ltd. (Laboratory of the Government Chemist, Teddington, UK), an international anti-doping certification organization.

*Cordyceps militaris* mycelium extraction powder (300 mg) was contained in each capsule so that the intake amount was 1800 mg per day in 6 capsules. In a previous study, the erythropoietin level was increased at a dose of 25 mg/kg of *Cordyceps sinensis* in mice [11]. Thus, the intake amount was calculated to be 1500 mg in humans, based on the body weight of the subjects, which was approximately 60 kg. In this study, the sufficient intake amount was set at 1800 mg per day. The placebo food was also produced by La Vie En Sante Ltd. The test food capsule (containing 300 mg of *Cordyceps militaris* mycelium extraction powder, 6.8 mg of silicon dioxide, 17 mg of calcium stearate, and 16.2 mg of potato starch) and the placebo food capsule (containing approximately 300 mg of potato starch) had the same appearance so were indistinguishable from each other.

### 2.4. Blood Tests

Blood tests were performed at 6:40 AM. After waking up, the subjects were allowed to drink only water, asked to skip breakfast and instructed to walk, not run, to the laboratory. The measured blood test markers are as follows: red blood cell (normal range, 427–500 × 10^4^/μL), hemoglobin (13.5–17.6 mg/dL), hematocrit (39.8–51.8%), serum iron (54–200 μg/dL), transferrin saturation index (TSAT; Fe ÷ TIBC × 100) (>20%), unsaturated iron-binding capacity (UIBC; 104–259 μg/dL), total iron-binding capacity (TIBC; 253–365 μg/dL), ferritin (39.4–340 ng/mL), erythropoietin (4.2–23.7 mIU/mL), white blood cell (3900–9800/μL), total protein (6.7–8.3 g/dL), triglyceride (50–149 mg/dL), total cholesterol (150–219 mg/dL), HDL cholesterol (40–86 mg/dL), creatine kinase (62–287 U/L), γ-glutamyl transpeptidase (GT; <70 IU/L), aspartate aminotransferase (AST; 10–40 U/L), alanine aminotransferase (ALT; 5–40 U/L), C-reacting protein (<0.14 mg/dL), urea nitrogen (8–22 mg/dL), uric acid (3.7–7.0 mg/dL), creatinine (0.61–1.04 mg/dL), testosterone (1.31–8.71 ng/mL), sodium (136–147 mEq/L), potassium 3.5–5.0 mEq/L), calcium (8.5–10.2 mg/dL), magnesium (1.8–2.6 mg/dL) and chloride (98–109 mEq/L). The blood tests were performed by SRL, Inc. (Tokyo, Japan).

### 2.5. Statistical Analysis

The data were shown as mean ± SD. In the case of hemoglobin, hematocrit, ferritin and erythropoietin, the change rates were calculated by comparing the values at 4 weeks, 8 weeks and 16 weeks after the test food intake with those at registration (0 week) and expressed as percentage; the change in the values at 4 weeks, 8 weeks and 16 weeks were divided by those at 0 week and multiplied by 100. Since some datasets did not meet the assumptions of normality, non-parametric methods were used for the statistical analysis. The Shapiro-Wilk test was used to determine normality in this study. The Wilcoxon signed–rank test or the Friedman test (*p* values adjusted by Bonferroni) was used for the intra-group comparison, and the Mann–Whitney U test was used for comparisons between the two independent (placebo and CM) groups. Statistical analysis was performed using SPSS software version 29 (IBM Corporation, Chicago, IL, USA). The significance level was set at *p* < 0.05. Since this study was designed to evaluate the effect of the CM intake, the authors did not set a primary endpoint, and did not perform the power analysis beforehand.

## 3. Results

### 3.1. Background Characteristics of the Subjects during Study Period

Monthly running distances of the subjects from August to November were 320~490 km in the placebo and CM groups, and the total running distances of the subjects were 1732.0 ± 23.9 km in the placebo group and 1623.5 ± 369.1 km in the CM group (Table 2). There were no significant differences between the two groups in the monthly running distances and total running distances. Regarding the 5000 m running time, the record time was significantly shortened after the test food intake (November 2019) compared with that at the registration (July 2019) in both the placebo and CM groups (*p* < 0.001); however, there was no significant difference between the two groups (Table 3). Moreover, the amounts of dietary intake of iron and vitamin C, which affects the absorption of iron in the gastrointestinal tract, were compared between the placebo and CM groups during study period based on the dietary survey using BDHQ. The results indicated that there was no significant difference in the dietary intake of iron and vitamin C between the two groups at all time points during the study period (0 week, at the registration; 4 weeks, 8 weeks, 12 weeks and 16 weeks after the test food intake) (Table 4). Furthermore, no adverse health effects attributable to the test food intake were observed during 16 weeks of the study period.

The official record for the 5000 m run were measured at registration (pre-intervention), and after finishing the test food intake (post-intervention). However, two subjects (one each in the placebo group and the CM group) did not participate in the 5000 m run conducted after the intervention. Therefore, those two subjects were excluded from the 5000 m record. Thus, the values of the placebo (*n* = 10) and CM (*n* = 10) groups are shown as the mean ± SD, and compared between pre- and post-intervention.

### 3.2. Blood Test Markers Related to Oxygen Transport Capacity

The levels of red blood cell, hemoglobin, hematocrit and TIBC did not essentially change during the study period in both the placebo and CM groups (Table 5). In contrast, the levels of serum iron and transferrin saturation index (TSAT; Fe ÷ TIBC × 100) were decreased during the study period in both the placebo and CM groups; however, these changes were not significant during the study period or between the two groups. The level of UIBC was more increased in the CM group compared with that in the placebo group during the study period, and the level was significantly increased in the CM group at 16 weeks after the test food intake compared with that at registration (*p* < 0.05). Interestingly, the serum ferritin level was decreased in the placebo group but was increased in the CM group during the study period, and the level was significantly decreased in the placebo group at 8 weeks after the food intake compared with that at registration (*p* < 0.001) (Figure 2A; Table 5). Moreover, the erythropoietin levels were significantly increased in both the placebo and CM groups at 16 weeks after the food intake compared with those at registration (*p* < 0.05) (Figure 2B; Table 5). However, there were no significant differences between the two groups during the study period in relation to the levels of UIBC, serum ferritin and erythropoietin.

Furthermore, the change rates (%) in hemoglobin, hematocrit, ferritin and erythropoietin were calculated by comparing the values at 4 weeks, 8 weeks and 16 weeks with those at registration (0 week) (Figure 3). Importantly, the change rates of hemoglobin and hematocrit were significantly increased in the CM group compared with those in the placebo group at 8 weeks after the test food intake (*p* < 0.05) (Figure 3A,B). Moreover, the change rates of ferritin were increased in the CM group but decreased in the placebo group, and the levels were significantly decreased in the placebo group compared with those of the CM group at 4 weeks and 8 weeks after the test food intake (*p* < 0.05) (Figure 3C). In contrast, the change rates of erythropoietin were increased during the study period in both the placebo and CM groups, and there was no significant difference between the two groups at 4 weeks, 8 weeks and 16 weeks after the test food intake (Figure 3D).

### 3.3. Other Blood Test Markers

The level of white blood cells was significantly lower in the CM group than in the placebo group at registration and at 16 weeks after the test food intake (*p* < 0.05) (Table 5). Moreover, the white blood cell level was significantly decreased in the placebo group at 8 weeks during study period (*p* < 0.01). The level of C-reacting protein was significantly decreased at 8 weeks after the test food intake in the placebo group (*p* < 0.01) (Table 6). The urea nitrogen level was significantly decreased at 8 weeks and 16 weeks after the test food intake in both the placebo and CM groups (*p* < 0.05). The uric acid level was significantly decreased at 16 weeks after the test food intake in the placebo group (*p* < 0.05). In contrast, the triglyceride level was significantly increased at 16 weeks after the study food intake in both the placebo and CM groups (*p* < 0.05). However, there were no significant differences between the two groups during the study period in relation to the levels of C-reacting protein, urea nitrogen, uric acid and triglyceride. Regarding total protein, total cholesterol, HDL cholesterol, GT, AST, ALT, creatinine, testosterone, sodium, potassium, calcium, magnesium and chloride, there were no significant differences found during the study period (0 week, 4 weeks, 8 weeks and 16 weeks) in the placebo and CM groups, or between the two groups during the study period (Table 5 and Table 6).

Notably, the levels of creatine kinase were increased to above the normal range (62–287 U/L) in both the placebo and CM groups at registration (Table 5). Interestingly, the creatine kinase level did not significantly change in the placebo group during the test period; however, the level was significantly decreased in the CM group at 8 weeks after the test food intake (*p* < 0.05) (Table 5). Moreover, the creatine kinase level was significantly decreased in the CM group compared with that in the placebo group at 16 weeks after the test food intake (*p* < 0.05) (Figure 4, Table 5).

Except for creatine kinase, the abovementioned changes during the study period were within the normal range.

## 4. Discussion

In this study, to investigate the effect of *Cordyceps militaris* (CM) mycelium extract on blood markers for anemia, a placebo-controlled, double-blind, parallel-group comparison trial was conducted, involving test foods (a placebo food and a CM extract–containing food) being administered to long-distance runners for 16 weeks during the pre-season training period. The values of the following parameters measured in the subjects were not significantly different between the placebo and CM groups during the study period: body weight, BMI and body fat percentage (Table 1); monthly running distances and total running distances (Table 2); and the amounts of dietary intake of iron and vitamin C (Table 4). In contrast, the 5000 m running time was significantly shortened after the test food intake compared with that at registration in both the placebo and CM groups; however, there was no significant difference between the two groups (Table 3). Furthermore, there were no adverse health effects observed that were attributable to the test foods during the study period.

It is possible that ambient temperatures and seasonal changes may have affected the performance of the athletes, the hematological markers and other physiological indicators recorded in the log test period (16 weeks). However, we confirmed that the changes in the blood tests and hematological markers were within the normal range during the test period (Table 5 and Table 6). Moreover, the body measurement markers (body weight, body mass index, body fat percentage, skeletal muscle percentage and basal metabolic rate) were not essentially changed during the test periods (Table 1). In contrast, the performance of the athletes, as evaluated by their 5000 m running time, was significantly faster in both the placebo and CM groups owing to the pre-season training (Table 3). Therefore, we believe that the effects of ambient temperatures and seasonal changes were minimal on the hematological markers and physiological indicators during the log test period (16 weeks) in this study.

Regarding the effect of the *Cordyceps sinensis* supplement on exercise performance, consistent results were not obtained. Specifically, some studies have indicated that the *Cordyceps sinensis* supplement improves exercise performance in healthy older subjects [12] and aerobic performance in long-distance track and field athletes [7]. In contrast, other studies have shown that the *Cordyceps sinensis* supplement has no effect on aerobic exercise performance in physically inactive (sedentary) young adults [13] and endurance-trained cyclists [14,15,16]. In the present study, the 5000 m running time was significantly shortened in not only the CM group, but also the placebo group after the test food intake, and the administration of the *Cordyceps militaris* mycelium extract did not exhibit a distinctive effect on endurance performance (5000 m running) (Table 3), although the administration exhibited an improving effect on the markers for anemia and muscle injury in long-distance runners, as described below. However, it may be difficult to simply compare the effects of different *Cordyceps sinensis* supplement and *Cordyceps militaris* mycelium extract on the exercise performance.

The total amount of iron in the body is approximately 4–5 g; 65% functions as the heme of hemoglobin in red blood cells, 4% functions as the heme of myoglobin in muscle, and 15–30% is complexed with ferritin and stored in the liver and other organs [17,18]. Iron metabolism in the human body is a semi-closed circuit with no active excretion mechanism. Thus, only 1 to 2 mg of iron is lost per day in healthy individuals, through the mechanisms of sweat and the detachment of mucous membranes and epithelial cells [17,18]. Factors contributing to enhanced iron excretion include increased iron loss in sweat; increased excretion of red blood cells, hemoglobin and myoglobin in urine; and gastrointestinal bleeding [17,18]. Notably, the half-life of iron is shorter in long-distance runners than in non-athlete controls, suggesting that iron turnover is enhanced in runners [9,10]. In addition, iron demand is thought to increase in athletes, owing to increased blood volume and increased iron-containing enzymes and myoglobin in muscle [9,10]. Moreover, it is possible that gastrointestinal function is decreased because of intense training, which may reduce iron absorption from the intestines [9,10]. This study was conducted during pre-season training, in which long-distance runners perform intensive running training before the official season to improve their endurance capacity. The total distance covered by the long-distance runners participating in this study was more than 1600 km (from July to November) (Table 3), and their 5000 m running records were shortened in both the placebo and CM groups (Table 2). Thus, it is assumed that the endurance performance of the subjects was increased, and the iron loss and demand were possibly enhanced in the long-distance runners during the study period.

Iron deficiency anemia is the most common type of anemia [17,18]. Prolonged iron deficiency leads to a depletion in stored iron, followed by decreased serum iron levels, and finally resulting in iron deficiency anemia accompanied by reduced hemoglobin. The state in which stored iron is reduced is called latent iron deficiency, the state in which serum iron is reduced is labeled overt iron deficiency, and the state in which hemoglobin is reduced is identified as iron deficiency anemia [17,18]. Interestingly, endurance athletes experience a decreased exercise performance when their serum iron levels are low, even if they do not develop iron deficiency anemia [9,10]. The amount of iron stored in the body is reflected in the serum ferritin levels [19,20]. In this study, the serum ferritin level was slightly decreased in the placebo group during the study period and was significantly decreased at 8 weeks after the food intake compared with that at registration (*p* < 0.001) (Figure 2A; Table 5). In contrast, the serum ferritin level was moderately increased in the CM group during the study period, although there was no significant difference during the study period (0 week, 4 weeks, 8 weeks, 12 weeks and 16 weeks after the test food intake) (Figure 2A; Table 5). Moreover, the serum ferritin levels were not significantly different between the placebo and CM groups during the study period. Interestingly, however, when the change rates were evaluated during the study period, the serum ferritin levels were moderately increased in the CM group but decreased in the placebo group, and the levels were significantly increased in the CM group compared with those in the placebo group at 4 weeks and 8 weeks after the test food intake (*p* < 0.05) (Figure 3C). These results suggest that the test food containing *Cordyceps militaris* mycelium extract has the potential to suppress the decrease in iron storage (as evidence by the reduction in serum ferritin) in endurance athletes (long-distance runners).

In a healthy individual, the lifespan of red blood cells is approximately 120 days. Interestingly, the serum level of erythropoietin (a hematopoietic hormone) was significantly increased at 16 weeks after the test food intake compared with that at registration in both the placebo and CM groups (*p* < 0.05) (Figure 2B; Table 5). Moreover, when the change rates were evaluated during the study period, the erythropoietin levels were gradually increased at 8 weeks and 16 weeks after the test food intake in both the placebo and CM group, and there was no significant difference between the two groups during the study period (Figure 3D). These observations suggest that erythropoietin levels are likely increased in both the placebo and CM groups during the study period, possibly accompanied by the loss of iron in the long-distance runners (as demonstrated by the decrease in serum iron and TSAT). However, the intake of test food containing *Cordyceps militaris* mycelium extract did not exhibit a distinctive effect on erythropoietin levels in the long-distance runners.

Erythropoietin stimulates the production of red blood cells and increases hemoglobin and hematocrit levels [21]. In the present study, the levels of red blood cells, hemoglobin and hematocrit did not essentially change during the study period in both the placebo and test food groups (Table 5). However, when the change rates were evaluated during the study period, the levels of hemoglobin and hematocrit were significantly increased in the CM group compared with those in the placebo group at 8 weeks after the test food intake (*p* < 0.05) (Figure 3A,B). Moreover, ferritin levels were significantly increased in the CM group compared with those in the placebo group at 4 weeks and 8 weeks after the test food intake (*p* < 0.05) (Figure 3C). Based on these observations, it could be speculated that, although the erythropoietin levels were increased in both the placebo and CM groups, possibly accompanied by iron loss, the hemoglobin and hematocrit levels were increased or maintained in the CM group owing to the suppression of iron storage loss, whereas the hemoglobin and hematocrit levels were decreased in the placebo group owing to the decrease in iron storage (as demonstrated by the reduction in serum ferritin). Thus, the intake of the test food containing *Cordyceps militaris* mycelium extract is expected to effectively maintain the hemoglobin and hematocrit levels in long-distance runners by suppressing the decrease in iron storage during pre-season training.

The results of the present study indicate that the change rates in the levels of hemoglobin, hematocrit and ferritin significantly increased in the CM group compared with those in the placebo group, although the absolute values of these parameters were within the normal range and were not significantly different between the two groups during the test period. This is possibly based on the findings that the effect of *Cordyceps militaris* mycelium extract is minimal, just increasing the change rates of hemoglobin, hematocrit and ferritin, but not exceeding the normal range of these markers.

Intense physical exercise such as endurance running induces an inflammatory response in muscle, and results in muscle damage accompanied with an increase in serum creatine kinase (a biomarker for inflammatory response in muscle) [22,23,24]. In this context, it has been reported that the serum creatine kinase level is increased during or immediately after exercise in long-distance runners such as marathon runners [22,23,24]. Notably, in the present study, the serum creatine kinase levels were increased above the normal range (62–287 U/L) in both the placebo and CM groups at registration (Table 4), suggesting that inflammation and damage were induced in the muscle of the subjects participating in this study. Interestingly, the level of creatine kinase was significantly decreased in the CM group at 8 weeks after the test food intake (*p* < 0.05) (Figure 4, Table 5), and the level was significantly decreased in the CM group compared with that in the placebo group at 16 weeks after the test food intake (*p* < 0.05) (Figure 4, Table 5). Since *Cordyceps sinensis* contains substances with anti-inflammatory and antioxidant potential [1,2,3,4,25,26], it could be speculated that the *Cordyceps militaris* mycelium extract administered in this study exhibits anti-inflammatory and antioxidant actions on muscle, thereby protecting muscle injury in long-distance runners during pre-season training.

The results of the present study indicate that the level of C-reacting protein was significantly decreased at 8 weeks in the placebo group; however, the level was not essentially changed in the CM group. Since the administration of the *Cordyceps militaris* mycelium extract significantly decreased the level of CK (a biomarker for inflammatory response in muscle), it could be speculated that *Cordyceps militaris* mycelium extract exhibits anti-inflammatory action on injured muscle as evaluated by CK but not C-reacting protein.

This study has some limitations. First, when the change rates of hemoglobin and hematocrit were evaluated during the study period, these anemic markers were significantly increased in the CM group compared with those in the placebo group at 8 weeks after the test food intake (*p* < 0.05) (Figure 3A,B). However, the values of hemoglobin and hematocrit themselves were not significantly changed during the study period in both the placebo and test food groups (Table 5). This could be explained by the fact that the subjects participating in this study were long-distance runners belonging to the track and field team of a university, and they were well nourished without any evidence of anemia, making it difficult to determine the clear effect of the test food on the anemic markers (hemoglobin and hematocrit). To clarify the effect of the test food, studies should be conducted using subjects with a risk of anemia (such as uncontrolled amateur athletes). Second, the present study revealed the potential of *Cordyceps militaris* mycelium to suppress a decrease in iron storage (as demonstrated by a decrease in serum ferritin levels) in long-distance runners during the pre-season training period. However, the mechanism for the action of the test food is not clear. Moreover, it is unclear which components are responsible for the action of the *Cordyceps militaris* mycelium extract. Although the effects of Dongchong-hongxiacao (*Cordyceps)* on exercise performance have been mostly evaluated using supplements containing *Cordyceps sinensis* [6,7,8], in this study, we utilized extracts of *Cordyceps militaris* mycelium. *Cordyceps* contains various kinds of chemical constituents including nucleotides/nucleotide derivatives, polysaccharides, sterols and fatty acids, and other compounds [1]. Although the pharmacologically active components of *Cordyceps* are still unresolved, at least two chemical constituents, cordycepin and cordycepic acid have been identified and proposed as important active constituents [1]. Interestingly, cordycepin and cordycepic acid are contained in both *Cordyceps militaris* and *Cordyceps sinensis* as common components [1]. In the future, it will be necessary to elucidate the mechanism for the action of *Cordyceps militaris* mycelium extract and to identify the components of *Cordyceps militaris* mycelium that contribute to the suppression of muscle injury and the decrease in iron storage. Third, in the present study, the study population was limited to long-distance runners belonging to a university. In future, it would be interesting to evaluate the effect of *Cordyceps militaris* mycelium extract using athletes from other sports or even non-athletes.

## 5. Conclusions

The present study has shown that *Cordyceps militaris* mycelium extract is expected to effectively maintain the hemoglobin and hematocrit levels in long-distance runners by suppressing the decrease in iron storage during pre-season training. Moreover, the extract was found to alleviate muscle injury (as demonstrated by the decrease in serum creatine kinase levels) in long-distance runners, possibly owing to its anti-inflammatory and antioxidant potential.

## Figures and Tables

**Figure 1 nutrients-16-01835-f001:**
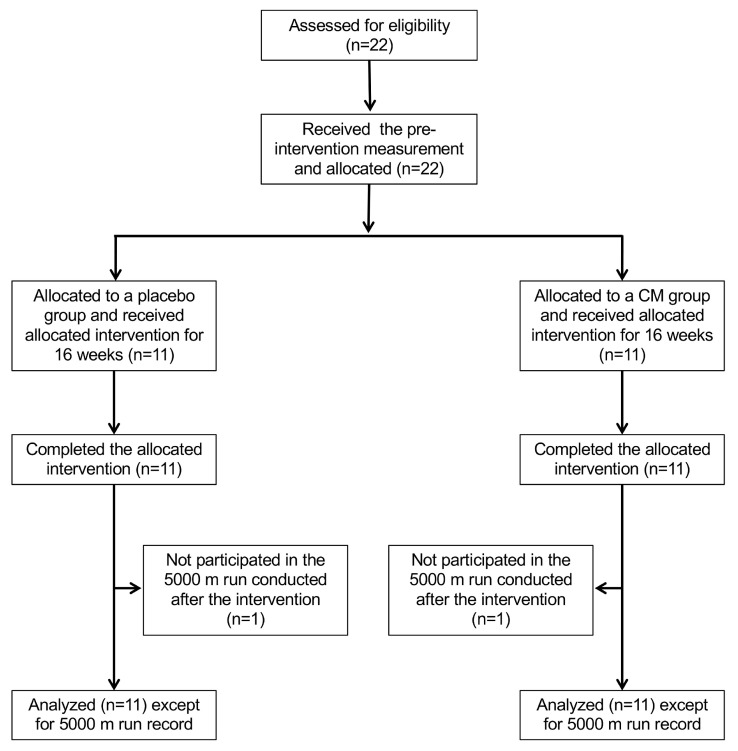
Flow diagram of the subjects who participated in the study.

**Figure 2 nutrients-16-01835-f002:**
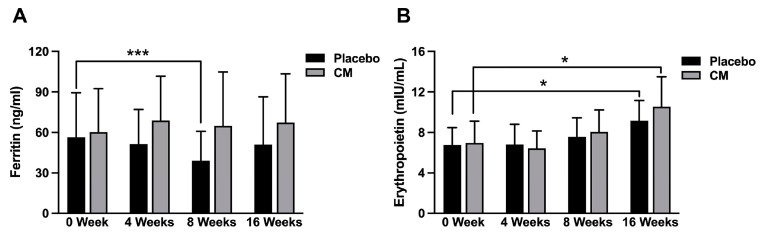
Serum levels of ferritin and erythropoietin in the placebo and CM groups during the study period. Serum levels of ferritin (**A**) and erythropoietin (**B**) were measured in the placebo (*n* = 11) and CM (*n* = 11) groups during the study period (0 week, at registration; 4 weeks, 8 weeks, 12 weeks and 16 weeks after the test food intake). Values are shown as the mean ± SD, comparing 0 week with 8 weeks, 12 weeks or 16 weeks. * *p* < 0.05, *** *p* < 0.001.

**Figure 3 nutrients-16-01835-f003:**
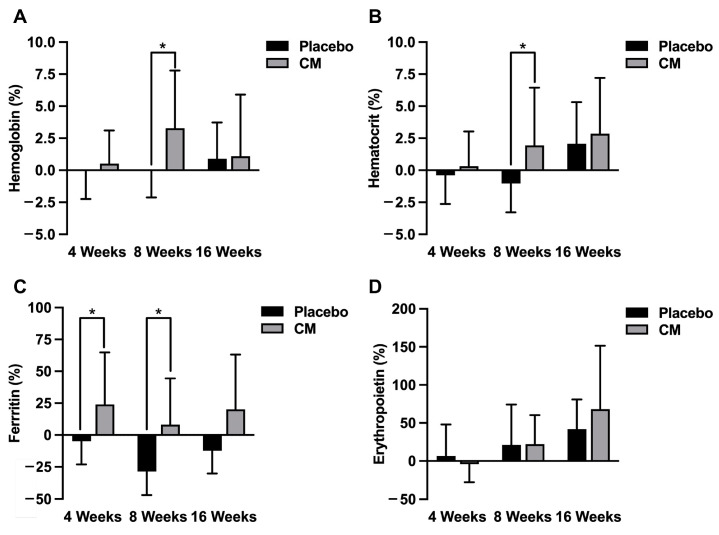
Change rates in the levels of hemoglobin, hematocrit, ferritin and erythropoietin in the placebo and CM groups during the study period. The change rates were calculated by comparing the values at 4 weeks, 8 weeks and 16 weeks after the test food intake with those at registration (0 week) in hemoglobin (**A**), hematocrit (**B**), ferritin (**C**) and erythropoietin (**D**), and expressed as a percentage. Data are shown as the mean ± SD, and compared between the placebo (*n* = 11) and CM (*n* = 11) groups. * *p* < 0.05.

**Figure 4 nutrients-16-01835-f004:**
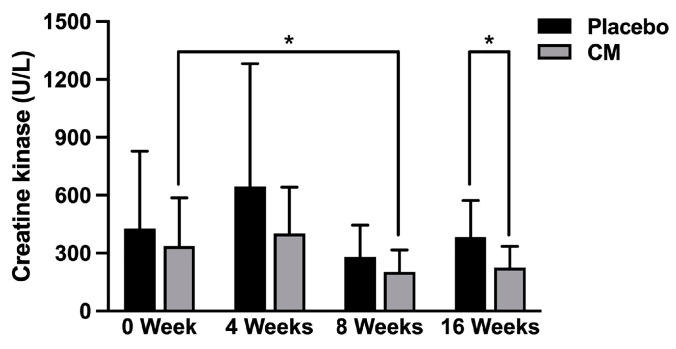
Serum level of serum creatin kinase in the placebo and CM groups during the study period. The serum levels of serum creatin kinase were measured in the placebo (*n* = 11) and CM (*n* = 11) groups during the study period (0 week, at registration; 4 weeks, 8 weeks, 12 weeks and 16 weeks after the test food intake). Values are shown as the mean ± SD, comparing 0 week with 8 weeks, 12 weeks or 16 weeks, and comparing the placebo and CM groups. * *p* < 0.05.

**Table 1 nutrients-16-01835-t001:** Body weight, body mass index, body fat percentage, skeletal muscle percentage and basal metabolic rate of the subjects in the placebo and CM groups during the study period.

	Groups	0 Week	4 Weeks	8 Weeks	16 Weeks
Body weight (kg)	Placebo	56.6 ± 4.4	56.4 ± 4.8	56.8 ± 4.8	57.3 ± 5.0
	CM	57.2 ± 4.3	57.0 ± 4.0	57.1 ± 3.8	57.7 ± 3.8
Body mass index (kg/m^2^)	Placebo	19.5 ± 0.9	19.4 ± 1.0	19.5 ± 1.1	19.7 ± 1.0
	CM	19.4 ± 1.3	19.3 ± 1.2	19.4 ± 1.2	19.6 ± 1.4
Body fat percentage (%)	Placebo	14.5 ± 2.1	14.0 ± 2.3	14.4 ± 2.5	14.4 ± 2.3
	CM	14.0 ± 2.5	13.7 ± 2.5	14.7 ± 2.7	14.0 ± 2.9
Skeletal muscle percentage (%)	Placebo	41.8 ± 1.4	42.3 ± 1.5	42.0 ± 1.6	42.1 ± 1.4
	CM	41.8 ± 1.3	42.1 ± 1.6	41.6 ± 1.6	42.0 ± 1.9
Basal metabolic rate (kcal)	Placebo	1452.6 ± 74.7	1451.6 ± 80.5	1455.7 ± 81.1	1466.1 ± 85.6
	CM	1463.7 ± 70.3	1461.2 ± 65.2	1459.9 ± 63.1	1473 4 ± 62.4

Values of the placebo (*n* = 11) and CM (*n* = 11) groups are shown as the mean ± SD.

**Table 2 nutrients-16-01835-t002:** Monthly running distance of the subjects in the placebo and CM groups during the study period.

Groups	August	September	October	November	Total
Placebo (km)	485.1 ± 127.5	432.0 ± 54.3	488.7 ± 44.3	326.2 ± 66.2	1732.0 ± 231.9
CM (km)	474.3 ± 107.5	369.8 ± 133.9	456.3 ± 115.2	323.0 ± 83.8	1623.5 ± 389.1

Values of the placebo (*n* = 11) and CM (*n* = 11) groups are shown as the mean ± SD.

**Table 3 nutrients-16-01835-t003:** Measurements of the 5000 m running time of the subjects in the placebo and CM groups before and after the intervention.

Groups	Pre-Intervention	Post-Intervention	*p*
Placebo (km)	15:18.82 ± 00:31.60	14:54.44 ± 00:32.00	<0.001
CM (km)	15:22.75 ± 00:35.31	14:55.98 ± 00:33.93	<0.001

**Table 4 nutrients-16-01835-t004:** Intake of dietary iron and vitamin C in the placebo and CM groups during the study period.

Dietary Components	Weeks	Placebo Group	CM Group
Iron (mg/1000 kcal)	0 week	4.35 ± 0.94	4.29 ± 0.87
	4 weeks	3.98 ± 0.85	3.78 ± 0.67
	8 weeks	4.10 ± 0.65	4.26 ± 1.01
	12 weeks	4.14 ± 0.68	3.80 ±0.65
	16 weeks	4.24 ± 0.76	4.19 ± 0.67
Vitamin C (mg/1000 cal)	0 week	60.55 ± 19.43	55.45 ± 16.08
	4 weeks	50.39 ± 23.94	48.55 ± 15.67
	8 weeks	50.35 ± 19.17	50.94 ± 21.06
	12 weeks	55.53 ± 23.78	54.58 ± 16.22
	16 weeks	55.45 ± 21.76	59.79 ± 20.51

Values of the placebo (*n* = 11) and CM (*n* = 11) groups are shown as the mean ± SD.

**Table 5 nutrients-16-01835-t005:** Changes in blood test markers in the subjects of the placebo and CM groups during the study period.

	Groups	0 Week	4 Weeks	8 Weeks	16 Weeks
Red blood cells (10^4^/mL)	Placebo	487.36 ± 40.26	487.36 ± 41.26	485.00 ± 38.61	492.82 ± 34.90
	CM	489.09 ± 35.00	491.73 ± 39.37	502.55 ± 37.50	495.64 ± 34.56
	Mann–Whitney U	NS	NS	NS	NS
Hemoglobin (g/dL)	Placebo	14.85 ± 1.14	14.83 ± 1.04	14.84 ± 1.14	14.96 ± 1.00
	CM	14.85 ± 0.79	14.92 ± 0.78	15.33 ± 0.98	15.00 ± 0.94
	Mann–Whitney U	NS	NS	NS	NS
Hematocrit (%)	Placebo	44.01 ± 3.08	43.81 ± 2.83	43.55 ± 3.01	44.87 ± 2.68
	CM	44.22 ± 1.90	44.36 ± 2.40	45.05 ± 2.30	45.46 ± 2.37
	Mann–Whitney U	NS	NS	NS	NS
Iron (mg/mL)	Placebo	112.45 ± 37.45	110.00 ± 24.42	110.00 ± 37.62	89.82 ± 25.96
	CM	128.64 ± 27.93	120.45 ± 23.66	127.82 ± 26.62	94.82 ± 33.94
	Mann–Whitney U	NS	NS	NS	NS
TSAT (%)	Placebo	34.28 ± 11.6	33.28 ± 7.6	32.22 ± 9.52	28.07 ± 7.97
	CM	40.29 ± 8.94	37.73 ± 8.77	38.62 ± 9.39	29.62 ± 9.49
	Mann–Whitney U	NS	NS	NS	NS
UIBC (mg/dL)	Placebo	217.45 ± 44.62	222.27 ± 35.40	226.82 ± 27.51	230.73 ± 32.35
	CM	192.82 ± 40.45	204.09 ± 48.76	206.36 ± 41.90	222.82 ± 34.03 *
	Mann–Whitney U	NS	NS	NS	NS
TIBC (mg/dL)	Placebo	329.91 ± 22.54	332.27 ± 28.02	336.82 ± 31.91	320.55 ± 22.78
	CM	321.45 ± 34.49	324.55 ± 39.83	334.18 ± 25.43	317.64 ± 37.10
	Mann–Whitney U	NS	NS	NS	NS
Ferritin (ng/mL)	Placebo	56.42 ± 33.06	51.46 ± 25.54	39.02 ± 21.79 ***	51.06 ± 35.43
	CM	60.28 ± 32.13	68.86 ± 32.80	64.89 ± 39.94	67.43 ± 36.15
	Mann–Whitney U	NS	NS	NS	NS
Erythropoietin (mIU/mL)	Placebo	6.75 ± 1.74	6.82 ± 1.99	7.58 ± 1.87	9.16 ± 1.98 *
	CM	6.98 ± 2.14	6.44 ± 1.71	8.05 ± 2.17	10.55 ± 2.96 *
	Mann–Whitney U	NS	NS	NS	NS
White blood cells (/mL)	Placebo	7127 ± 1861	6081 ± 1031	5100 ± 1037 **	5918 ± 643
	CM	5627 ± 716	5518 ± 753	5327 ± 752	5236 ± 452
	Mann–Whitney U	<0.05	NS	NS	<0.05
Total protein (g/dL)	Placebo	7.32 ± 0.35	7.34 ± 0.37	7.17 ± 0.33	7.14 ± 0.34
	CM	7.24 ± 0.30	7.30 ± 0.17	7.35 ± 0.38	7.19 ± 0.25
	Mann–Whitney U	NS	NS	NS	NS
Triglyceride (mg/dL)	Placebo	47.82 ± 29.23	41.36 ± 16.16	53.55 ± 19.59	82.09 ± 35.77 *
	CM	50.27 ± 21.74	47.55 ± 14.43	61.91 ± 29.87	77.91 ± 32.63 *
	Mann–Whitney U	NS	NS	NS	NS
Total cholesterol (mg/dL)	Placebo	182.27 ± 32.05	184.00 ± 33.37	185.64 ± 31.33	188.73 ± 29.07
	CM	172.27 ± 18.55	175.64 ± 19.61	177.64 ± 22.38	177.82 ± 22.95
	Mann–Whitney U	NS	NS	NS	NS
HDL cholesterol (mg/dL)	Placebo	77.36 ± 15.23	79.18 ± 14.30	80.64 ± 15.64	75.45 ± 13.49
	CM	68.73 ± 14.54	71.91 ± 12.96	71.18 ± 14.28	69.18 ± 14.93
	Mann–Whitney U	NS	NS	NS	NS
Creatine kinase (U/L)	Placebo	427.91 ± 399.60	645.73 ± 636.14	280.45 ± 164.85	384.64 ± 188.84
	CM	336.73 ± 249.79	402.00 ± 239.84	203.18 ± 113.12 *	225.82 ± 109.49
	Mann–Whitney U	NS	NS	NS	<0.05

Values of the placebo (*n* = 11) and CM (*n* = 11) groups are shown as the mean ± SD. The Mann–Whitney U test was used for comparisons between the placebo and CM groups. The Friedman test was used for the intra-group comparison during the study period (comparison between 0 week and 4 weeks, 8 weeks, 12 weeks or 16 weeks). * *p* < 0.05, ** *p* < 0.01, *** *p* < 0.001. NS, not significant.

**Table 6 nutrients-16-01835-t006:** Changes in blood test markers in the subjects of the placebo and CM groups during the study period.

	Groups	0 Week	4 Weeks	8 Weeks	16 Weeks
GT (IU/L)	Placebo	22.73 ± 5.35	24.55 ± 8.04	23.45 ± 7.93	22.00 ± 5.50
	CM	21.45 ± 5.65	24.73 ± 10.90	24.09 ± 8.13	21.09 ± 5.99
	Mann–Whitney U	NS	NS	NS	NS
AST (U/L)	Placebo	39.45 ± 22.15	46.00 ± 27.04	32.00 ± 10.14	38.55 ± 13.84
	CM	29.27 ± 8.91	33.27 ± 12.77	26.00 ± 6.26	28.55 ± 6.68
	Mann–Whitney U	NS	NS	NS	NS
ALT (U/L)	Placebo	7.32 ± 0.35	7.34 ± 0.37	7.17 ± 0.33	7.14 ± 0.34
	CM	7.24 ± 0.30	7.30 ± 0.17	7.35 ± 0.38	7.19 ± 0.25
	Mann–Whitney U	NS	NS	NS	NS
C-reacting protein (mg/dL)	Placebo	0.36 ± 0.89	0.05 ± 0.06	0.02 ± 0.01 **	0.05 ± 0.02
	CM	0.05 ± 0.03	0.04 ± 0.01	0.04 ± 0.01	0.06 ± 0.03
	Mann–Whitney U	NS	NS	NS	NS
Urea nitrogen (mg/dL)	Placebo	19.85 ± 5.93	18.51 ± 4.56	15.09 ± 2.99 ***	15.62 ± 3.41 **
	CM	18.92 ± 6.37	17.75 ± 5.12	14.32 ± 3.28 ***	15.21 ± 4.78 *
	Mann–Whitney U	NS	NS	NS	NS
Uric acid (mg/dL)	Placebo	5.78 ± 0.80	5.92 ± 0.95	5.51 ± 0.83	4.95 ± 0.66 *
	CM	5.85 ± 0.89	5.86 ± 0.65	5.34 ± 0.65	4.94 ± 0.78
	Mann–Whitney U	NS	NS	NS	NS
Creatinine (mg/dL)	Placebo	0.78 ± 0.08	0.78 ± 0.09	0.75 ± 0.07	0.78 ± 0.08
	CM	0.82 ± 0.06	0.83 ± 0.08	0.79 ± 0.06	0.80 ± 0.05
	Mann–Whitney U	NS	NS	NS	NS
Testosterone (ng/mL/L)	Placebo	6.25 ± 1.85	6.09 ± 2.14	6.05 ± 1.83	6.09 ± 1.95
	CM	7.24 ± 2.39	7.05 ± 2.33	7.12 ± 2.16	7.01 ± 1.76
	Mann–Whitney U	NS	NS	NS	NS
Sodium (mEq/L)	Placebo	140.36 ± 1.29	140.18 ± 0.87	140.00 ± 1.00	140.91 ± 0.94
	CM	141.18 ± 1.17	140.73 ± 1.27	140.45 ± 1.04	141.27 ± 1.35
	Mann–Whitney U	NS	NS	NS	NS
Potassium (mEq/L)	Placebo	4.35 ± 0.41	4.17 ± 0.28	4.14 ± 0.29	4.32 ± 0.24
	CM	4.12 ± 0.15	4.15 ± 0.29	4.15 ± 0.20	4.39 ± 0.34
	Mann–Whitney U	NS	NS	NS	NS
Calcium (mg/dL)	Placebo	9.41 ± 0.23	9.35 ± 0.25	9.45 ± 0.23	9.41 ± 0.21
	CM	9.41 ± 0.20	9.44 ± 0.31	9.52 ± 0.20	9.51 ± 0.29
	Mann–Whitney U	NS	NS	NS	NS
Magnesium (mg/dL)	Placebo	2.31 ± 0.11	2.30 ± 0.13	2.23 ± 0.13	2.32 ± 0.10
	CM	2.25 ± 0.09	2.26 ± 0.12	2.23 ± 0.12	2.29 ± 0.11
	Mann–Whitney U	NS	NS	NS	NS
Chloride (mEq/L)	Placebo	103.27 ± 1.10	103.64 ± 0.92	103.09 ± 1.30	103.45 ± 1.13
	CM	103.55 ± 1.21	103.00 ± 1.34	103.00 ± 1.57	103.82 ± 1.25
	Mann–Whitney U	NS	NS	NS	NS

Values of the placebo (*n* = 11) and CM (*n* = 11) groups are shown as the mean ± SD. The Mann–Whitney U test was used for comparisons between the placebo and CM groups. The Friedman test was used for the intra-group comparison during the study period (comparison between 0 week and 4 weeks, 8 weeks, 12 weeks or 16 weeks). * *p* < 0.05, ** *p* < 0.01, *** *p* < 0.001. NS, not significant.

## Data Availability

Data are contained within the article.

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
