# Peer review of "Effect of the Administration of Cordyceps militaris Mycelium Extract on Blood Markers for Anemia in Long-Distance Runners"

_nutrients, 2024, doi:10.3390/nu16121835_

Round 1

Reviewer 1 Report

Comments and Suggestions for Authors

The article utilized Cordyceps militaris mycelium extract as an interventional modality to investigate its effects on hematological indices associated with anemia in a cohort of long-distance runners. The findings suggest that it may exert a protective influence against muscle injury. However,the specific components of the Cordyceps militaris mycelium extract that contribute to its efficacy, as well as the mechanisms underlying its protective effects against muscle damage need further in-depth exploration."

Issue 1: The authors used a placebo-controlled, double-blind, parallel- 19 group comparison trial. However, a 16-week exercise intervention was conducted for a longer period of time, and the effects of ambient temperature and seasonal changes on the performance of the athletes, Hb and other physiological indicators were not mentioned.

Issue 2: The results showed that Cordyceps militaris mycelium extract had a positive effect on hemoglobin and hematocrit, but the authors also noted that these metrics did not change significantly in either the placebo or experimental groups. Further exploration is needed to determine why the observed rates of change were significantly different between the two groups, while the absolute values were not.

Issue 3: In this study, CK was used to determine muscle damage and inflammation in order to hypothesize that Cordyceps militaris mycelium extract has anti-inflammatory and antioxidant effects on muscle. Based on this, other serum markers such as CRP and TNF-α can be introduced for further mechanistic studies.

Issue 4:The long-term safety of Cordyceps militaris mycelium extract intake requires further evaluation. Additionally, the study population was comprised of long-distance runners from the university. It is recommended that the authors discuss the generalizability of their findings to the general population or to athletes of different categories."

Comments on the Quality of English Language

The arrangement of words and phrases to create well-formed sentences must be improved.

Reviewer 2 Report

Comments and Suggestions for Authors

Nakamura et al. provides interesting insights into how Cordyceps militaris Mycelium extract influences blood parameters for anemia in long-distance runners. Throughout the entire manuscript, however, there are numerous grammatical, sentence structure, and syntax errors that make it difficult to read. Additionally, the entirety of the introduction does not sufficiently rationalize the purpose of the current study, wherein the authors begin by mentioning one fungus while then in the methods supporting another. While my specific recommendations are outlined below, the existing manuscript does provides a comprehensive analysis on how Cordyceps militaris Mycelium extract may potentially help runners alleviate the onset of anemia. Before acceptance, however, major revisions are essential, and one recommendation is to enlist the support of a colleague who is proficient in English grammar to help alleviate some of the concerns listed below.

General Comments:

Introduction:

The manuscript contains numerous grammatical errors, significantly hindering comprehension and obscuring the authors' intended message. For instance, "The host of fungus is the larvae of Chinese moth (Hepialus armoricanns Oben)" should be corrected to "The host of the fungus is the larvae of the Chinese moth (Hepialus armoricanns Oben)." Another example is the sentence, “When the larva is infected by Ophiocordyceps sinensis in winter, fungus slowly grows inside the larva, utilizes the nutrients in the larva, and emerges from the ground in summer with the distinctive shape of Dong chong hong xia cao retaining the larva in the ground" is overly complex and incredibly awkward to read.

Moreover, the manuscript includes extraneous information that does not support the study's rationale or hypothesis, further complicating adequate interpretation. For example, the detailed explanation of the name "TOCHU-KASO" and its history in Japan is not directly relevant to the study's hypothesis and can be condensed or omitted. The passage —"In Japan, the name 'TOCHU-KASO' (winter worm summer grass) has been used for over 200 years as the insect bearing fungi, and there are approximately 400 species worldwide and about 300 species in Japan for the fungi of Dong chong hong xia cao "— could easily be streamlined to maintain focus on the study's main points.

A thorough revision focusing on correcting grammatical errors, improving sentence structure, and enhancing overall syntax is essential. Simplifying the language to ensure clarity and precision, and removing unnecessary details that do not contribute directly to the study's rationale, are critical steps. The introduction should follow a logical sequence, guiding the reader from the background and rationale through to the specific research question and hypothesis. Each paragraph should build on the previous one, contributing to a coherent and compelling narrative. For instance, the sentence "Previously, it has been revealed that the components of Cordyceps sinensis have variety of activities including the suppression of cancer cell growth, the improvement of quality of life (QOL) in cancer patients, the amelioration of liver dysfunctions, the alleviation of coldness and malaise in perimenopausal women, and the reduction in low-density lipoprotein (LDL) cholesterol in the elderly" could be revised to remove unnecessary wording.

Materials and Methods:

Similar to the introduction, the methods are riddled with grammatical errors, sentence structure, and overall syntax. Another glaring issue is that the entire rationale -albeit weak- is supporting the use of Cordyceps sinensis extract, but in the methods the focus is on another completely different fungus. This creates an expectation that the study will focus on this particular fungal extract. However, as one delves into the methods, it becomes apparent that the procedures described revolve around a completely different fungus. This disconnect between the introduction and methods not only confuses readers but also calls into question the validity and purpose of the research. Moreover, it is almost impossible or rather incredibly difficult to assess how the extract was administered as it is not clearly stated while only brief mentions that the subjects consumed either placebo or Cordyceps militaris foods.

For the statistical analysis section, why was there no power analysis done? Moreover, did the authors choose to measure the normality of all their results as they only mentioned that some results did not meet normality assumptions. Please provide more information regarding what variables were assessed as well as how they were assessed.

Results

The results section reads almost identical to a discussion, which significantly compromises the clarity and structure of the paper. It is inundated with jargon and explanations that belong more appropriately in the discussion or methods sections, making it difficult for readers to discern the actual findings from their interpretations. This section should focus solely on presenting data clearly and concisely, without delving into explanations or interpretations. The excessive jargon and detailed explanations obscure the actual results, while methodological details should be confined to the methods section.

Specific comments:

Page 1, Lines 43-44: Insert a comma after “…(winter worm summer grass)…”. Additionally, there needs to be an “a” between “with” and “club-shaped”. Once done the sentence should be the following: “…(winter worm summer grass), is an entomopathogenic fungus with a club-shaped…).

Page 1, Line 45: Please change “altitude” into “altitudes”.

Page 2, Lines 50-61: Please combine these paragraphs into one as there is no reason for these to be separated.

Page 2, Line 53: Please remove the comma after Ophiocordyceps synesis.

Page 2, Line 60: Remove just the word “cholesterol”.

Page 2, Lines 62-63: Condense this sentence as there is a lot of unnecessary information. Specifically, change 10 healthy male subjects into 10 healthy untrained males and remove, “who did not usually exercise”. Moreover,

Page 2, Line 65-66: Correct “was” into “were” for both instances. Additionally, this sentence is rather unclear as to what was being administered. As a reader, you can assume its Cordyceps sinensis mycelium that is being administered but the current structure makes it rather unclear.

Page 2, Line 81: Although TOCHU-KASO is another name for the Cordyceps sinensis, please keep the name unanimous throughout the entirety of the manuscript to not cause any confusion.

Page 2, Line 84: Why are we now referring to Cordyceps militaris when the entire introduction focused on a different fungus? Please explain.

Page 3, Line 109: Is Cordyceps militaris the same as Cordyceps sinensis mycelium or a different fungus altogether? Please explain.

Comments on the Quality of English Language

As mentioned previously, please consider enlisting the assistance of a native english speaker.

Round 2

Reviewer 1 Report

Comments and Suggestions for Authors

The article utilized Cordyceps militaris mycelium extract as an interventional modality to investigate its effects on hematological indices associated with anemia in a cohort of long-distance runners. The authors have revised and explained the comments point-by- point. However,the difference  pharmacologic effect between Cordyceps sinensis extracts and Cordyceps militaris mycelium need further in-depth exploration.

Issue 1: The pharmacological effect of Cordyceps sinensis extracts and Cordyceps militaris mycelium is very different. The Cordyceps sinensis mycelium can not completely replace cordyceps sinensis extract. The author should make  some corresponding supplementary explanations in the manuscript.

Issue 2: In the discussion part Line 480-490, it is inappropriate for the author to compare cordyceps sinensis extract. with Cordyceps sinensis mycelium . It should be discussed the effects produced by the Cordyceps pharmacological ingredients.

Comments on the Quality of English Language

The quality of English writing basically meets the requirements of publication.

Reviewer 2 Report

Comments and Suggestions for Authors

The authors have sufficiently addressed my concerns and should be commended for their efforts.

Author Response

The authors would like to thank Reviewer 2 for the careful review and appropriate evaluation of this manuscript.